# Control of MXenes' electronic properties through termination and intercalation

James L. Hart [1], Kanit Hantanasirisakul [1,2], Andrew C. Lang[1], Babak Anasori[1,2], David Pinto[1,2], Yevheniy Pivak[3], J. Tijn van Omme [3], Steven J. May[1], Yury Gogotsi[1,2] & Mitra L. Taheri[1]

MXenes are an emerging family of highly-conductive 2D materials which have demonstrated state-of-the-art performance in electromagnetic interference shielding, chemical sensing, and energy storage. To further improve performance, there is a need to increase MXenes' electronic conductivity. Tailoring the MXene surface chemistry could achieve this goal, as density functional theory predicts that surface terminations strongly influence MXenes' Fermi level density of states and thereby MXenes' electronic conductivity. Here, we directly correlate MXene surface de-functionalization with increased electronic conductivity through in situ vacuum annealing, electrical biasing, and spectroscopic analysis within the transmission electron microscope. Furthermore, we show that intercalation can induce transitions between metallic and semiconductor-like transport (transitions from a positive to negative temperature-dependence of resistance) through inter-flake effects. These findings lay the groundwork for intercalation- and termination-engineered MXenes, which promise improved electronic conductivity and could lead to the realization of semiconducting, magnetic, and topologically insulating MXenes.

[1] Department of Materials Science & Engineering, Drexel University, Philadelphia, PA 19104, USA. [2] A.J. Drexel Nanomaterials Institute, Drexel University, Philadelphia, PA 19104, USA. [3] DENSsolutions, Informaticalaan 12, Delft 2626ZD, The Netherlands. Correspondence and requests for materials should be addressed to M.L.T. (email: mtaheri@coe.drexel.edu)

Discovered in 2011, MXenes are a rapidly growing family of 2D transition metal carbides, nitrides, and carbonitrides with the general formula $M_{n+1}X_nT_x$ ($n = 1$, 2, or 3; M = transition metal, e.g., Ti, V, Nb, Mo; X = C and/or N; T = surface termination, e.g., –OH, –F, =O)[1–5]. MXenes are formed by selective etching parent ternary carbide MAX compounds to remove the A-group element, e.g., $Ti_3AlC_2$ (layered MAX) → $Ti_3C_2T_x$ (2D MXene)[6,7]. For both device applications and fundamental studies, MXene samples are generally thin films comprised of many MXene flakes, though some studies have focused on single-layer MXene[8,9]. In contrast to most other 2D materials, MXenes offer an attractive combination of high electronic conductivity, hydrophilicity, and chemical stability[1–5,10–13]. With these properties, MXenes show exceptional promise in areas including electromagnetic interference shielding[14,15], wireless communication[16], chemical sensing[17–20], energy storage[21–23], optoelectronics[24–27], triboelectrics[28–30], catalysis[31–33], and conformal/wearable electronics[34]. Performance in these applications is directly related to electronic conductivity, and as such, there is motivation to further increase the MXene metallic conductivity. In parallel, there is excitement surrounding the potential realization of semiconducting MXenes, which are predicted to be excellent materials for spintronics and thermoelectrics[35–37].

To meet these demands on MXenes' electronic properties, researchers have mostly focused on the development of new $M_{n+1}X_n$ chemistries. So far, over 30 MXenes have been synthesized, but the first discovered MXene—$Ti_3C_2T_x$—remains the most conductive[13]. Recently, certain Mo- and V-based MXenes have garnered interest due to their so-called semiconductor-like behavior[38–40], i.e., a negative temperature-dependence of resistance ($dR/dT$). However, the cause of negative $dR/dT$ in these MXenes remains unclear, and their underlying electronic structure is debated[38,41]. A potentially more useful approach to control MXenes' conductivity is to manipulate their surface chemistry. Surface terminations, which are introduced during MXene synthesis[6], have been predicted to control metal-to-insulator transitions[37,38] and to affect functional properties such as magnetism[1,42], Li-ion capacity[43], catalytic performance[31,44], band alignment[45], mechanical properties[46], and predictions of superconductivity[47]. While promising, these predicted effects lack direct experimental confirmation[48,49]. An additional mechanism which can affect MXene conductivity is intercalation. Intercalants are not thought to alter the intra-flake (intrinsic) MXene properties, but for multi-layer samples, intercalation can increase device resistance by over an order of magnitude[27,40,41,50–52]. This effect is generally attributed to intercalants increasing the inter-flake spacing and thereby the inter-flake resistance.

For any multi-layer MXene sample, intercalation, termination, and $M_{n+1}X_n$ chemistry all contribute to the measured electronic conductivity, and this convolution of effects greatly complicates experimental interpretation. As a result, our understanding and ability to control MXenes' electronic properties are lacking. To address this challenge, we perform in situ vacuum annealing (up to 775 °C) and electrical biasing of MXenes within the transmission electron microscope (TEM). We observe de-intercalation and surface de-functionalization with in situ electron energy loss spectroscopy (EELS) and ex situ thermogravimetric analysis with mass spectroscopy (TGA-MS). Importantly, we utilize low-dose direct detection (DD) EELS[53] to avoid electron beam-induced specimen damage[54] (Supplementary Figure 1). With this approach, we correlate the desorption of –OH, –F, and =O termination species with increased MXene conductivity. Additionally, we report transitions from ensemble semiconductor-like (negative $dR/dT$) to metallic behavior in $Ti_3CNT_x$ and $Mo_2TiC_2T_x$ after the de-intercalation of water and organic molecules. This work furthers our fundamental understanding of conduction through MXene films and opens the door to intercalation- and termination-engineered MXenes.

## Results

**Sample synthesis and experimental approach.** We investigated three MXenes with diverse macroscopic electronic transport behavior: $Ti_3C_2T_x$, $Ti_3CNT_x$, and $Mo_2TiC_2T_x$ (Table 1). $Ti_3C_2T_x$ (its structure is shown schematically in the Fig. 1a top inset) is the most studied MXene and is known to be metallic and highly conductive[7]. Density functional theory (DFT) studies have consistently predicted that surface functionalization reduces the $Ti_3C_2T_x$ density of states (DOS) at the Fermi level ($E_F$), suggesting a decrease in the charge carrier density and thereby a decrease in the conductivity[43,55,56]. Understanding of the next two MXenes—$Ti_3CNT_x$ and $Mo_2TiC_2T_x$—is limited. The structure of $Ti_3CNT_x$ is similar to $Ti_3C_2T_x$ but with a mixture of C and N on the X-sites (Fig. 1b top inset). DFT predicts $Ti_3CNT_x$ to be metallic for all terminations[55,57,58], but to date, this MXene has only shown semiconductor-like transport (unpublished results). $Mo_2TiC_2T_x$ is an ordered, double transition metal MXene analogous to $Ti_3C_2T_x$ but with the outer Ti layers replaced by Mo layers[59] (Fig. 1c top inset). Terminations have been predicted to induce metallic, semiconducting[38], and topologically insulating[60] states in this MXene. Experimentally, $Mo_2TiC_2T_x$ shows semiconductor-like behavior in its as-prepared state[38], but it is unclear if this behavior is due to intrinsic ($M_{n+1}X_nT_x$ stoichiometry) or extrinsic (intercalation, inter-flake hopping) effects.

We produced the $Ti_3C_2T_x$ and $Ti_3CNT_x$ samples through etching of their 3D parent MAX phases, i.e., $Ti_3AlC_2$ and $Ti_3AlCN$, in a mixture of LiF and HCl. This process results in –OH, –F, and =O terminations and $H_2O$ and $Li^+$ intercalation[6,23]. $Mo_2TiC_2T_x$ was instead produced through etching of $Mo_2TiAlC_2$ in HF and delaminating via tetrabutylammonium hydroxide (TBAOH) intercalation[59]. This method reduces the concentration of –F termination and results in tetrabutylammonium ($TBA^+$) and $H_2O$ intercalation[40,61]. $TBA^+$ is a large organic ion which can significantly increase the inter-flake spacing and electrical resistance[6,40,52]. To directly compare $H_2O$ and $TBA^+$ intercalation, we additionally studied $Ti_3CNT_x$

---

**Table 1 Summary of in situ heating and biasing results**

| MXene chemistry | Termination | Intercalation | dR/dT | Resistance (Ω) |
|---|---|---|---|---|
| $Ti_3C_2T_x$ | $(OH)_{0.4}F_{0.4}O_{0.5} \rightarrow F_{0.2}O_{0.5}$ | $H_2O$, $Li^+ \rightarrow Li^+$ | M → M | 41 → 10 |
| $Ti_3CNT_x$ | $(OH)_{0.9}F_{0.5}O_{0.7} \rightarrow F_{0.2}O_{0.7}$ | $H_2O$, $Li^+ \rightarrow Li^+$ | S → M | 159 → 27 |
| $Ti_3CNT_x(TBA^+)$ | $(OH)_{0.6}F_{0.1}O_{1.2} \rightarrow O_{1.2}$ | $H_2O$, $TBA^+ \rightarrow$ none | S → S | 3330 → 290 |
| $Mo_2TiC_2T_x$ | $(OH)_{0.5}F_{0.01}O_{1.5} \rightarrow O_{0.8}$ | $H_2O$, $TBA^+ \rightarrow$ none | S → M | 2500 → 387 |

In each cell, the arrow symbol represents changes induced during in situ vacuum annealing up to temperatures of 775, 700, 750, and 775 °C for $Ti_3C_2T_x$, $Ti_3CNT_x$, $Ti_3CNT_x(TBA^+)$, and $Mo_2TiC_2T_x$, respectively. In the $dR/dT$ column, M indicates metallic conduction (positive $dR/dT$) while S indicates semiconductor-like conduction (negative $dR/dT$). Initial termination concentrations were determined through ex situ X-ray photoelectron spectroscopy (XPS) measurements (Supplementary Tables 3-6 and Supplementary Figure 12). The loss of intercalants and –OH terminations was determined from ex situ TGA-MS, and the reduction in –F and =O terminating species was measured with in situ EELS

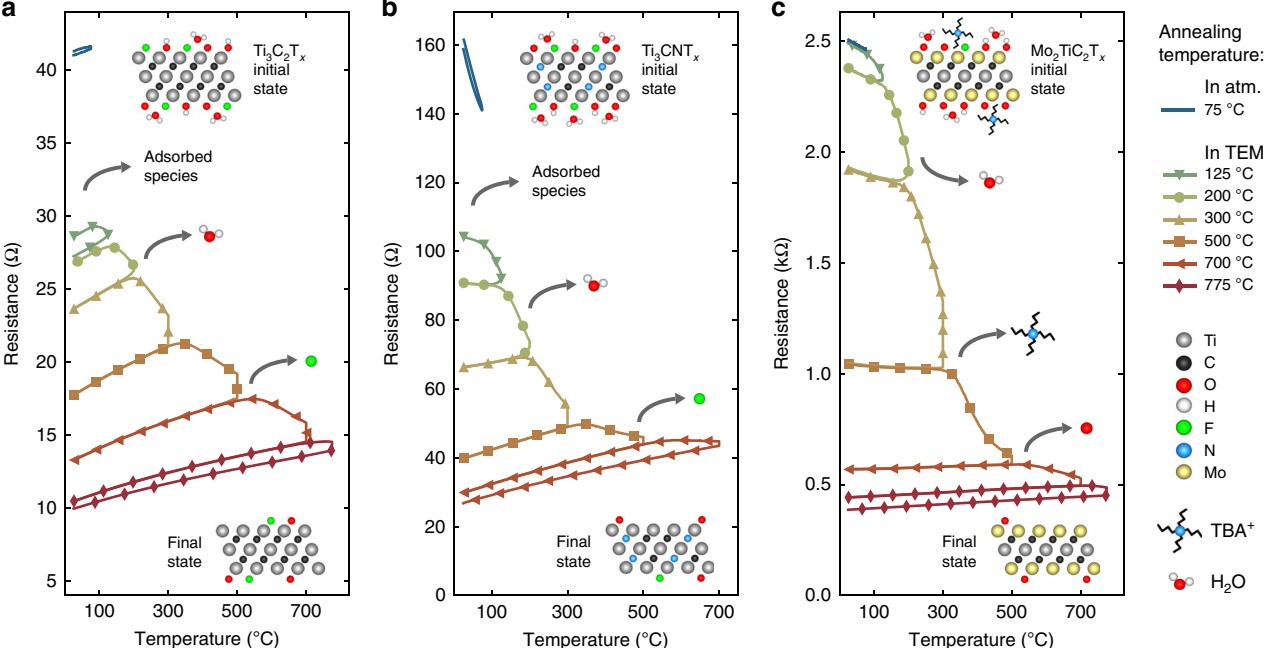

**Fig. 1** Evolution of MXene electronic properties with in situ vacuum annealing. Resistance versus temperature measurements are shown for $Ti_3C_2T_x$ (**a**), $Ti_3CNT_x$ (**b**), and $Mo_2TiC_2T_x$ (**c**). The measurements were conducted during various vacuum annealing steps performed in the TEM. Each vacuum annealing step is represented with a different color and symbol. For each annealing step, both the heating and cooling curves are shown. In all cases, the resistance decreased during annealing, hence, the cooling curve is always beneath the heating curve. The atomic structures of the various MXenes are shown as insets. The initial state schematics (top schematics) show $Ti_3C_2T_x$ and $Ti_3CNT_x$ with intercalated water molecules on their surfaces, and $Mo_2TiC_2T_x$ is shown with water and $TBA^+$ molecules. With annealing, the MXene sample resistance is affected by the loss of adsorbed species, intercalants, and terminating species. Some of these processes are shown schematically. The resistance data in this figure are also shown in Supplementary Figures 13-15, where the resistance is plotted as a function of annealing time

prepared with HF etching and TBAOH delamination. For clarity, this sample is referred to as $Ti_3CNT_x(TBA^+)$.

After synthesis, MXene films were spray-casted[24] onto MEMS (microelectromechanical systems)-based nanochips[62] designed for heating and biasing within the TEM column (Supplementary Figure 2). The MXene films were at least several flakes thick (Supplementary Figures 1 and 2), and the electrode spacing (~20 µm) was considerably larger than the MXene flake diameters (~100 nm up to 2–3 µm). As such, sample resistance measurements were dependent upon both intra- and inter-flake contributions. The cross-sectional sample areas were not well-defined, so we report the relative change in sample resistance and not the absolute resistivity. TEM imaging and electron diffraction confirmed the MXene structure both before and after in situ annealing at ≥700 °C (Supplementary Figure 2).

The temperature-dependent resistance of each sample was initially measured in ambient atmosphere from room temperature (RT) up to 75 °C in order to understand the as-prepared sample properties. In agreement with previous results, $Ti_3C_2T_x$ displayed metallic behavior while $Ti_3CNT_x$ and $Mo_2TiC_2T_x$ displayed semiconductor-like (negative $dR/dT$) behavior[27,38] (Table 1 and Fig. 1). After measurement of the as-prepared MXene electronic properties, the samples were inserted into the TEM and vacuum annealed at temperatures up to 775 °C.

Before providing a detailed analysis of our in situ heating and biasing experiments, we first summarize our two main findings. First, we observed transitions from semiconductor-like to metallic behavior in $Ti_3CNT_x$ and $Mo_2TiC_2T_x$ after the annealing-induced loss of intercalated species. These transitions reveal the intra-flake metallicity of $Ti_3CNT_x$ and $Mo_2TiC_2T_x$ and demonstrate that intercalants can cause negative $dR/dT$ in multi-layer MXenes. Second, high temperature annealing and the partial loss of surface

terminations (Fig. 1, bottom insets) increased the conductivity of all three MXenes. This finding is in agreement with past predictions that non-terminated MXenes exhibit metallic behavior with high carrier concentrations[43,55,56,59]. In the following, we describe the in situ heating and biasing data in greater detail; results are organized based on the different mechanisms which influence MXene electronic properties.

**Adsorbed species**. Prior to thermal annealing, insertion of $Ti_3C_2T_x$ and $Ti_3CNT_x$ into the TEM vacuum (~$10^{-5}$ Pa) caused an immediate reduction in resistance (Fig. 1a, b and Supplementary Figure 3). For both of these samples, the resistance decreased roughly 20% after 150 s of insertion into the TEM. This change in resistance is attributed to the loss of adsorbed atmospheric species, e.g., $H_2O$ and $O_2$. These species are known to cause doping in 2D materials such as graphene[63], and similar effects have previously been reported in $Ti_3C_2T_x$[8,25]. We note that both of these samples were only intercalated with $H_2O$ and $Li^+$. For $Mo_2TiC_2T_x$ intercalated with $TBA^+$, the sample resistance did not significantly change upon exposure to the TEM vacuum (Fig. 1c and Supplementary Figure 3). This observation suggests an increased effect of the large $TBA^+$ molecule relative to $H_2O$, specifically, that $TBA^+$ intercalation limits the film resistance and masks the effect of atmospheric species desorption on the electronic resistance. The effect of adsorbed species on the resistance of $Ti_3CNT_x(TBA^+)$ could not be determined, since the sample resistance prior to in situ vacuum annealing was too high to be accurately measured.

**Intercalation**. For all studied MXenes, de-intercalation significantly increased electronic conductivity. We first consider the

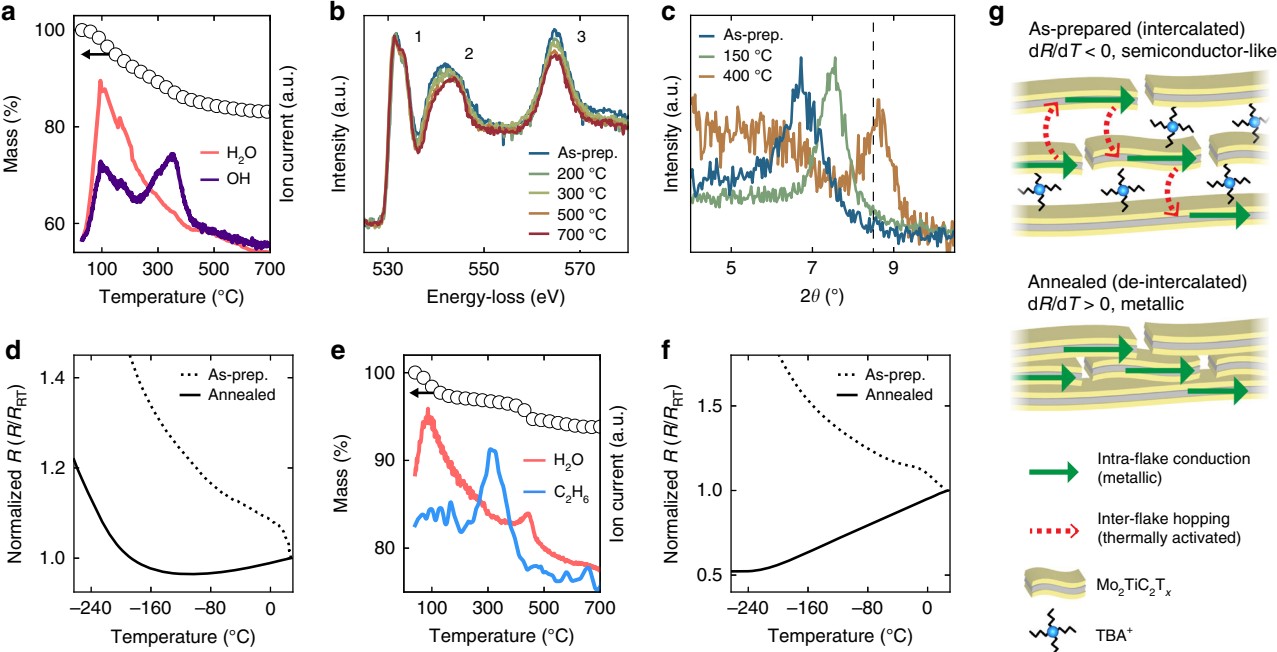

**Fig. 2** Influence of intercalated species on the MXene resistance and $dR/dT$. **a** TGA-MS of $Ti_3CNT_x$ showing the loss of $H_2O$ intercalants and –OH termination species. **b** In situ EELS of the normalized O $K$-edge of $Ti_3CNT_x$ indicating the loss of $H_2O$ intercalants with annealing. **c** Ex situ XRD of the $Ti_3CNT_x$ (002) peak for an as-prepared thick sample, i.e., vacuum filtered and free-standing, as well as thick samples annealed at 150 °C (in vacuum) and 400 °C (in Ar). The peak shift to higher $2\theta$ values represents a decrease in the inter-layer spacing. The dashed vertical line represents the approximate position of the $Ti_3C_2T_x$ (002) peak after annealing at 150 °C, taken from ref. [6]. **d** Temperature-dependent PPMS resistance measurements of a thick as-prepared (intercalated) $Ti_3CNT_x$ sample and the de-intercalated sample annealed to 700 °C within the TEM. $R_{RT}$ is the room temperature resistance. **e** TGA-MS of $Mo_2TiC_2T_x$ showing the loss of $H_2O$ intercalants and the decomposition of $TBA^+$ intercalants. **f** Temperature-dependent PPMS resistance measurements of a thick as-prepared (intercalated) $Mo_2TiC_2T_x$ sample and the de-intercalated sample annealed to 775 °C within the TEM. The as-prepared data is taken from ref. [38]. **g** Schematic of intercalants' influence on conduction through multi-flake $Mo_2TiC_2T_x$

role of $H_2O$, which began to de-intercalate at lower temperatures than $TBA^+$. For all samples, mass spectroscopy (MS) showed a large peak in the $H_2O$ (mass to charge ratio $(m/e) = 18$) ion current centered at ~150 °C, indicating $H_2O$ de-intercalation. This behavior is shown for $Ti_3CNT_x$ in Fig. 2a, and full TGA-MS results for all samples are presented in Supplementary Figure 4. In situ EELS data is consistent with the de-intercalation of $H_2O$ during low temperature annealing. Figure 2b shows the O $K$-edge of $Ti_3CNT_x$ after various in situ annealing steps; three distinct peaks are observed. For $=O$ or –OH bonded with surface Ti, these three peaks are expected, with peak 1 arising due to hybridization between the termination moiety and the Ti $3d$ orbitals[64]. Conversely, the O $K$-edge of water does not contain peak 1[65]. With annealing, peaks 2 and 3 decrease with respect to peak 1, signifying the loss of intercalated $H_2O$. The loss of $=O$ or –OH terminations cannot explain the observed changes in fine structure, as surface de-functionalization would produce a more uniform decrease in the O $K$-edge intensity.

By annealing samples at 200 °C, we isolate the influence of $H_2O$ de-intercalation on the $Ti_3C_2T_x$ and $Ti_3CNT_x$ electronic properties from any effects of surface de-functionalization. To prove this point, we consider the de-functionalization of –OH, the least stable termination species and thus the first to desorb[12,66]. The MS –OH $(m/e = 17)$ ion current in Fig. 2a shows two peaks for $Ti_3CNT_x$. The first peak perfectly mirrors the $H_2O$ ion current and is thus attributed to de-protonated $H_2O$ (Supplementary Figure 4); the second peak at ~375 °C is attributed to –OH termination loss. Our conclusion that –OH terminations are stable at <200 °C is supported by nuclear magnetic resonance spectroscopy and neutron scattering studies performed previously on $Ti_3C_2T_x$[67,68], assuming that

the termination interaction with surface Ti atoms is similar in both carbide and carbonitride MXenes. Moreover, it was shown that hydroxyl groups do not desorb from titania powder until 350–500 °C[69].

For 200 °C annealed $Ti_3C_2T_x$, $H_2O$ de-intercalation caused an 18% reduction in sample resistance, in qualitative agreement with past results (Fig. 1a)[50,70]. Subtle increases in the $dR/dT$ of $Ti_3C_2T_x$ with 125 and 200 °C annealing suggest the decrease of an insulating inter-flake resistive term, as opposed to a change in the intra-flake metallic conductivity (Supplementary Note 1, Supplementary Figure 5, and Supplementary Table 1). These changes in $dR/dT$ with low temperature annealing are in agreement with a decrease in resistance driven by $H_2O$ de-intercalation.

In comparison to $Ti_3C_2T_x$, $Ti_3CNT_x$ displayed an anomalously large response to $H_2O$ de-intercalation. After annealing at 200 °C, there was a 36% decrease in resistance and a transition from negative to positive $dR/dT$ (Fig. 1b). Ex situ XRD shows a contraction of the $Ti_3CNT_x$ $c$-lattice parameter (signifying a decrease in the inter-layer spacing) after 150 °C annealing (Fig. 2c). This measurement further confirms the partial de-intercalation of water and supports the argument that de-intercalation improves conductivity through a reduced inter-flake resistance.

The measurement of metallic conductivity in $Ti_3CNT_x$ confirms numerous theoretical predictions of its metallic nature[55,57,58]. To better understand the electronic properties of the de-intercalated $Ti_3CNT_x$, we studied the sample's low-temperature conduction ex situ. To do so, the sample was further annealed to 700 °C (ensuring maximum de-intercalation), removed from the TEM, and then inserted into the physical property measurement system (PPMS). Figure 2d shows the

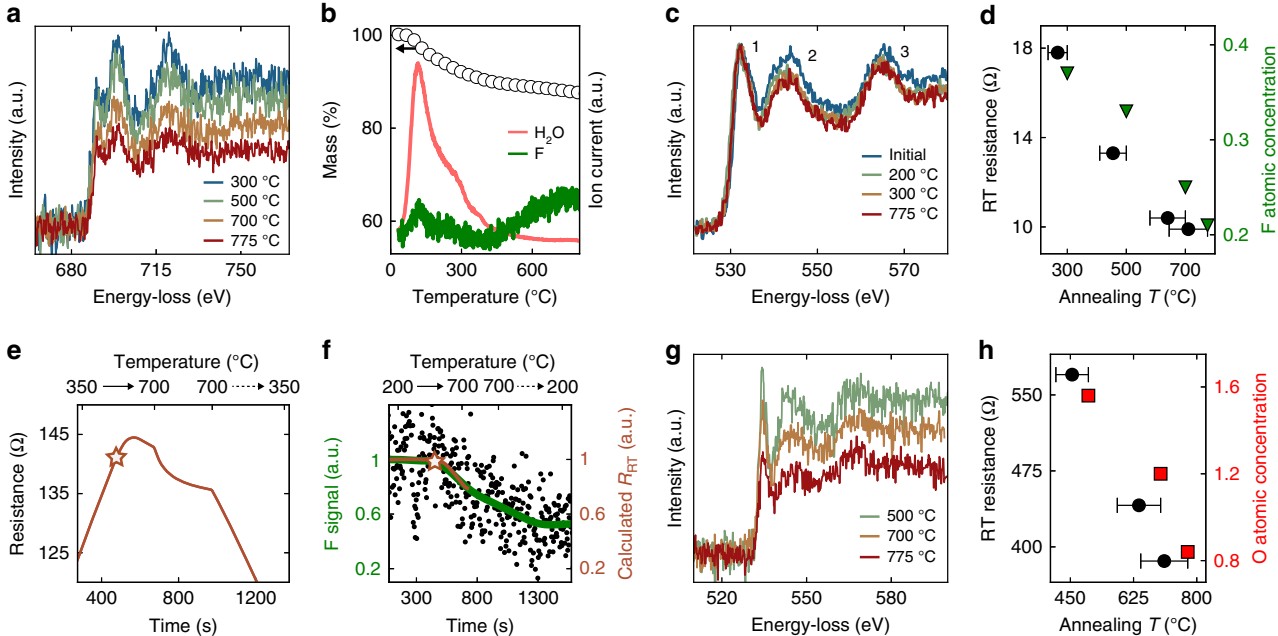

**Fig. 3** Correlation of MXene surface termination and conductivity. **a** In situ EELS of the Ti$_3$C$_2$T$_x$ F $K$-edge showing the loss of –F with annealing. **b** TGA-MS of Ti$_3$C$_2$T$_x$ showing de-intercalation of H$_2$O completing by ~400 °C and the onset of –F de-functionalization at ~400 °C. **c** In situ EELS of the Ti$_3$C$_2$T$_x$ O $K$-edge with annealing. The relative decrease in peaks 2 and 3 relative to peak 1 indicates the loss of intercalated H$_2$O and the retention of =O. **d** Comparison of the post-anneal RT resistance of Ti$_3$C$_2$T$_x$ (black circles) and –F concentration (green triangles) given in units of $x$ based on the chemical formula Ti$_3$C$_2$O$_{0.5}$F$_x$. **e**, **f** show the time-resolved resistance and F $K$-edge intensity, respectively, of Ti$_3$CNT$_x$ during in situ annealing. The solid (dashed) arrows represent heating (cooling). For the time axis, $t = 0$ s corresponds to the beginning of annealing at RT. The stars mark 500 °C, the maximum temperature of prior annealing. The green line in **f** is a Gaussian smooth of the EELS data (black circles). The dark orange line in **f** is the calculated RT resistance, determined using the measured value of d$R$/d$T$ upon heating. Because d$R$/d$T$ changes with annealing, this calculation is only valid up to ~700 °C. To highlight the correlation between the –F concentration and the calculated RT resistance, both quantities were normalized by their initial value. **g** In situ EELS spectra of the O $K$-edge of Mo$_2$TiC$_2$T$_x$. **h** Comparison of the post-anneal RT resistance (black circles) with the concentration of =O (red squares), given in units of $x$ based on the chemical formula Mo$_2$TiC$_2$O$_x$. For **a**, **c**, and **g**, EELS spectra were acquired at RT after annealing, and the edge intensities were normalized to the Ti $L$-edge intensity. In **d** and **h**, the change in elemental concentration was determined with in situ EELS. The temperature offset between elemental concentration data and resistance measurements in **d** and **h** are due to thermal gradients within the nanochip (Supplementary Figure 10). The range of these thermal gradients are represented by the horizontal error bars in the RT resistance data

normalized resistance of as-prepared (intercalated) and 700 °C annealed (de-intercalated) Ti$_3$CNT$_x$. The intercalated sample shows semiconductor-like behavior across the entire temperature range of the PPMS measurement (from RT to −263 °C), while the de-intercalated sample displays metallic behavior down to roughly −150 °C. At this temperature, the de-intercalated Ti$_3$CNT$_x$ sample shows a transition to negative d$R$/d$T$, which is similar to the reported behavior of multi-layer Ti$_3$C$_2$T$_x$ samples[27,38]. It is an open question whether this low temperature transition is due to intra-flake or inter-flake effects. Both the as-prepared and annealed samples displayed negative magnetore-sistance (MR) at −263 °C (10 K) (Supplementary Figure 6). We note that annealing Ti$_3$CNT$_x$ above 300 °C causes partial surface de-functionalization, which we discuss in the next section. However, high temperature annealing and surface de-functionalization did not significantly affect d$R$/d$T$ (Fig. 1b and Supplementary Table 1), so we ascribe the changes in normalized resistance shown Fig. 2d to de-intercalation.

While H$_2$O intercalation was known to increase MXene resistance values, the findings shown in Figs. 1a, b and 2a–d demonstrate that H$_2$O intercalation can additionally act to decrease the value of d$R$/d$T$. Moreover, given a sufficient concentration of intercalants in Ti$_3$CNT$_x$, the inter-flake resistance can mask the MXene's intra-flake metallic conductivity. We stress that the resulting negative d$R$/d$T$ of intercalated Ti$_3$CNT$_x$ is not due to a bandgap opening, rather it is the

consequence of the temperature-dependence of the inter-flake electron hopping process.

It is noteworthy that H$_2$O intercalated Ti$_3$C$_2$T$_x$ displayed metallic conduction while H$_2$O intercalated Ti$_3$CNT$_x$ displayed semiconductor-like conduction, despite both MXenes being intrinsically metallic. To understand this difference, we show the position of the Ti$_3$C$_2$T$_x$ (002) peak after annealing at 150 °C (vertical dashed line in Fig. 2c). The $c$-lattice spacing of 150 °C-annealed Ti$_3$CNT$_x$ (23.0 Å) is significantly larger than that of 150 °C-annealed Ti$_3$C$_2$T$_x$ (20.6 Å). The lattice spacing of Ti$_3$CNT$_x$ only contracts to ~20 Å after annealing at 400 °C. This data suggests an increased lattice response of Ti$_3$CNT$_x$ to intercalated H$_2$O, which in turn increases the inter-flake resistance and drives the transition to negative d$R$/d$T$.

Measurements of Mo$_2$TiC$_2$T$_x$, having both H$_2$O and TBA$^+$ intercalants, showed an increased effect of TBA$^+$ relative to H$_2$O. Annealing Mo$_2$TiC$_2$T$_x$ at 200 °C led to the de-intercalation of H$_2$O and a 24% reduction in resistance, but there was no change in the sign of d$R$/d$T$ (Figs. 1c and 2e). Subsequent annealing up to 500 °C led to a 69% decrease in resistance and a transition from negative to positive d$R$/d$T$ (Fig. 1c). This transition to metallic behavior is due to the decomposition and loss of TBA$^+$, as seen in the MS signal of C$_2$H$_6$ ($m/e = 30$) (Fig. 2e). The measured temperature of TBA$^+$ de-intercalation/decomposition was ~350 °C, which is similar to the decomposition temperature of TBA$^+$ intercalated in graphite[71]. Previous work has shown that annealing

$Mo_2TiC_2T_x$ at 530 °C reduces the $c$-lattice parameter from 37.7 to 24.5 Å[52], which supports our claim that the decomposition of $TBA^+$ causes a decrease in the inter-flake resistance and consequently a transition to metallic behavior.

To further study the effect of $TBA^+$ decomposition, the low-temperature resistance of $Mo_2TiC_2T_x$ was measured. We annealed the sample up to 775 °C in an attempt to eliminate residual $TBA^+$, and then we removed the sample from the TEM and inserted it into the PPMS. The normalized resistance of the as-prepared (intercalated) and the annealed (de-intercalated) $Mo_2TiC_2T_x$ are shown in Fig. 2f. The de-intercalated $Mo_2TiC_2T_x$ sample shows metallic behavior down to the lowest measured temperature of −263 °C (10 K), and both the as-prepared and de-intercalated samples show positive MR (Supplementary Figure 6). These results demonstrate the intrinsically metallic nature of $Mo_2TiC_2T_x$, in agreement with the previously reported metallic conductivity in this MXene after annealing at ~530 °C[52]. We note that annealing $Mo_2TiC_2T_x$ at 775 °C leads to the partial loss of termination groups; however, annealing at these high temperatures did not significantly alter the value of d$R$/d$T$ (Fig. 1c and Supplementary Table 1). Thus, the difference in normalized resistance shown in Fig. 2f is attributed to de-intercalation.

The observation of intercalation-induced negative d$R$/d$T$ in two different MXenes with two different intercalant species provides strong evidence that this is a general phenomenon. For sufficient levels of intercalation, the thermally activated inter-flake hopping process becomes the rate-limiting step for conduction through multi-layer samples. This extrinsic effect causes MXenes to display negative d$R$/d$T$ regardless of their intrinsic electronic properties, as shown schematically in Fig. 2g.

Next, we consider $Ti_3CNT_x$ intercalated with $TBA^+$. With annealing, $Ti_3CNT_x(TBA^+)$ showed a larger total reduction in resistance than $Ti_3CNT_x$ intercalated with only $H_2O$ and $Li^+$ (Table 1 and Supplementary Figure 7), consistent with the increased effect of $TBA^+$ intercalants. However, even after annealing at 750 °C, $Ti_3CNT_x(TBA^+)$ continued to display negative d$R$/d$T$, albeit with a reduced temperature-dependency (Supplementary Table 1). Given the previous demonstration of metallic intra-flake conduction in $Ti_3CNT_x$, we speculate that incomplete removal of $TBA^+$ is responsible for the persistence of the semiconductor-like behavior. After annealing, the temperature-dependent resistance closely followed $R \propto T \times \exp(W/kT)$, which may be related to the inter-flake hopping mechanism (Supplementary Figure 7). It is presently unclear why annealing led to metallic conduction in $Mo_2TiC_2T_x(TBA^+)$ but not in $Ti_3CNT_x(TBA^+)$. This difference could be due to the surface Mo atoms of $Mo_2TiC_2T_x$ affecting the release of $TBA^+$ and/or the inter-flake hopping process.

**Termination**. After de-intercalation, vacuum annealing at higher temperatures led to surface de-functionalization and improved electronic conductivity. Initially, each sample had −OH, −F, and =O terminations in varying concentrations (Table 1). Based on DFT calculations, −OH should be the first species to desorb[12,66], and for certain samples, TGA-MS data indicated the loss of −OH ($m/e = 17$) at ~375 °C (Fig. 2a). However, for the majority of TGA-MS measurements, the release of de-protonated $H_2O$ completely masked the signal of −OH loss (Supplementary Figure 4). De-functionalization of −OH was also difficult to detect with in situ EELS (Fig. 2b), which could be due to the electron beam transforming −OH groups into more stable =O terminations prior to EELS acquisition[72]. Conversely, the partial loss of −F and =O species were clearly identified. As such, we focus on the effects of −F and =O desorption; however, we stress that the release of these species indicates prior loss of less stable −OH groups[12,66].

Annealing $Ti_3C_2T_x$ from 300 to 775 °C led to a significant loss of −F termination species and an associated increase in sample conductivity. In situ EELS measurements show a decrease in the F $K$-edge intensity after annealing at 500, 700, and 775 °C (Fig. 3a), demonstrating the partial de-functionalization of −F. Consistent with the EELS measurements, ex situ TGA-MS shows the release of −F termination ($m/e = 19$) beginning around 400 °C (Fig. 3b). The broad peak in the $m/e = 19$ ion channel at ~150 °C closely mirrors the $H_2O$ channel, indicating that this peak arises from a mechanism related to $H_2O$ de-intercalation, e.g., release of protonated $H_2O$. Despite the large loss of −F, there was no evidence of =O release, as demonstrated by the constant value of peak 1 of the O $K$-edge EELS spectra (Fig. 3c).

The observed loss of −F beginning at ~400 °C and the retention of =O terminations is consistent with ref. [73], where Persson et al. report in situ annealing of $Ti_3C_2T_x$ with scanning TEM (STEM) and XPS analysis. However, our results differ from the recent in situ STEM annealing of $Ti_3C_2T_x$ reported by Sang et al.[54]. In their study, =O terminations were lost and large voids were formed after 500 °C annealing and electron irradiation. We attribute these contradictory results to beam induced effects. Sang et al. utilized a focused STEM probe (~$10^9$ e$^-$ s$^{-1}$ Å$^{-2}$) while we used a spread TEM beam (~10 e$^-$ s$^{-1}$ Å$^{-2}$). Moreover, our use of low-dose DD EELS[53] allows short acquisition times, further decreasing the potential for beam-induced sample degradation (Supplementary Figure 1).

In Fig. 3d, we show the correlation between changes in the $Ti_3C_2T_x$ −F concentration and the RT resistance as a function of annealing temperature. The decrease in sample resistance produced during the 500 °C annealing step likely has a significant contribution from the loss of −OH groups. Additionally, there may be a small contribution from the final stages of $H_2O$ de-intercalation. However, after the 500 °C annealing step, ex situ TGA-MS (Fig. 3b) and in situ EELS (Fig. 3c) show no evidence of further $H_2O$ loss. Additionally, XRD studies have shown that there is no change in the $Ti_3C_2T_x$ $c$-lattice spacing for annealing beyond 500 °C[74]. With no evidence of intercalation loss, no change in the inter-flake spacing, and no observation of a conducting secondary phase (Supplementary Figure 2)[75], we conclude that the improvement in $Ti_3C_2T_x$ conductivity during annealing steps at 700 and 775 °C is entirely due to the loss of −F terminations.

Similar to $Ti_3C_2T_x$, in situ annealing of $Ti_3CNT_x$ at 500 and 700 °C led to a reduction in −F species and improved conductivity (Fig. 1b and Supplementary Figure 8). In addition to the $Ti_3CNT_x$ sample discussed previously in the text and shown in Figs. 1 and 2, another $Ti_3CNT_x$ sample was measured with time-resolved in situ EELS. The time-resolved measurements of $Ti_3CNT_x$ show the simultaneous release of −F termination species alongside an increase in electronic conductivity (Fig. 3e, f and Supplementary Figure 9). This data further demonstrates that the loss of termination species influences the MXene electronic properties.

For $Mo_2TiC_2T_x$, surface de-functionalization occurred at lower temperatures than $Ti_3C_2T_x$ or $Ti_3CNT_x$, indicating a weaker interaction between termination species and surface Mo atoms compared to Ti atoms. In situ EELS shows the complete loss of −F from $Mo_2TiC_2T_x$ after annealing at 500 °C (Supplementary Figure 8), and annealing at 700 and 775 °C produced a large reduction in the =O concentration (Fig. 3g). As =O terminations were partially removed from the surface, the $Mo_2TiC_2T_x$ resistance decreased a total of 32% (Fig. 3h). These findings suggest that the release of =O from $Mo_2TiC_2O_x$ increases the MXene intra-flake conductivity[38,59]. However, we cannot rule out the possibility that inter-flake effects contribute to the measured changes in resistance for annealing steps at 700 and 775 °C. As opposed to $H_2O$ intercalants which cleanly de-intercalate, the $TBA^+$ intercalants

decompose (Fig. 2e and Supplementary Figure 4). Even though the decomposition of $TBA^+$ leads to the onset of metallic behavior in $Mo_2TiC_2T_x$ (Fig. 1c), it is possible that some residue remains even after annealing at >500 °C. This is likely the case for $Ti_3CNT_x(TBA^+)$, since this MXene does not show metallic behavior even after annealing at 750 °C.

To the best of our knowledge, these results constitute the first direct experimental correlation of MXene surface chemistry and electronic conductivity. Previous DFT studies have predicted that termination of $Ti_3C_2T_x$ and $Ti_3CNT_x$ with –OH, –F and/or =O significantly alters the electronic states near $E_F$[7,43,55,56,66]. For non-terminated $Ti_3C_2$ and $Ti_3CN$, DFT predicts a local maximum in the DOS at $E_F$, but with complete surface functionalization, the $DOS(E_F)$ is greatly reduced. Consequently, surface functionalization may alter the electronic resistance through a decrease the carrier concentration, $n$. To test the validity of this proposed mechanism, we analyzed the concurrent changes in the $Ti_3C_2T_x$ and $Ti_3CNT_x$ resistance and $dR/dT$ with in situ annealing. We assume that the metallic intra-flake conductivity of these MXenes can be described by the Drude equation, and that over the temperature range of our in situ TEM experiments, electron-phonon scattering is approximately linear in temperature (Fig. 1)[76,77]. With these assumptions, the Drude equation predicts that an increase in $n$ with annealing will produce a proportional decrease in the resistance and $dR/dT$, i.e., $\Delta R \propto \Delta dR/dT \propto (n_1/n_2 - 1)$, where $n_1$ and $n_2$ are the carrier concentrations before and after annealing, respectively (Supplementary Note 1). To visualize this behavior, we define $\eta$ as the ratio of the proportional change in the RT $dR/dT$ to the proportional change in the RT resistance for a given annealing step. A value of $\eta = 1$ is predicted for a change in resistance driven solely by a change in the intra-flake carrier concentration. For annealing $Ti_3C_2T_x$ and $Ti_3CNT_x$ at high temperatures, $\eta \sim 1$, supporting the claim that surface de-functionalization increases the MXene conductivity through an increase in $n$ (Fig. 4).

In contrast to the $\eta \sim 1$ behavior for high temperature annealing, $\eta$ is negative for annealing $Ti_3C_2T_x$ and $Ti_3CNT_x$ at low temperatures ($\eta < 0$ reflects a decrease in resistance and an increase in $dR/dT$). As we describe in Supplementary Note 1, a negative value of $\eta$ is inconsistent with a change to the intra-flake metallic conductivity driven by, e.g., a change in $n$, effective electron mass, or defect density. However, a negative value of $\eta$ can be explained by a decrease in the insulating inter-flake resistance, assuming that the inter-flake and intra-flake resistances act in series. Thus, the observed transition from negative to positive $\eta$ is indicative of a transition from control over the inter-flake resistance (due to low temperature annealing and de-intercalation) to control over the intra-flake resistance (due to high temperature annealing and surface de-functionalization). For $Mo_2TiC_2T_x$ and $Ti_3CNT_x(TBA^+)$, $\eta < 0$ for all annealing temperatures, suggesting that residue from the $TBA^+$ decomposition continues to affect the MXene inter-flake resistance even after annealing at >500 °C (Supplementary Figure 5).

## Discussion

In this study, we vacuum annealed multi-layer MXene samples within the TEM and measured 4, 6, >10, and 6 times increases in the conductivity of $Ti_3C_2T_x$, $Ti_3CNT_x$, $Ti_3CNT_x$ ($TBA^+$), and $Mo_2TiC_2T_x$, respectively. With annealing, we studied de-intercalation and surface de-functionalization with both in situ and ex situ spectroscopic techniques. By correlating this chemical analysis with in situ resistance and $dR/dT$ measurements, we were able to delineate the effects of intercalation and surface termination on MXene electronic properties.

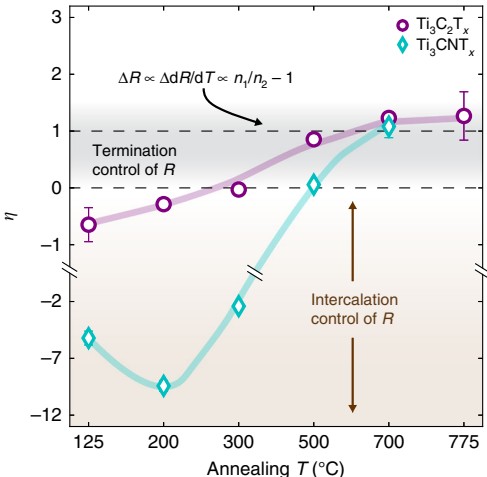

**Fig. 4** Analysis of concurrent resistance and $dR/dT$ changes with annealing. For a given annealing step, $\eta$ is the ratio of the proportional change in the RT $dR/dT$ to the proportional change in the RT resistance (Supplementary Note 1). A positive value of $\eta$ (indicating a decrease in both the $dR/dT$ and the resistance) is consistent with a change in the intra-flake resistance, and a negative value of $\eta$ (indicating an increase in $dR/dT$ and a decrease in resistance) is consistent with a change to the inter-flake resistance. For a change in resistance solely due to a change in the intra-flake carrier concentration, the Drude formula predicts that $\eta = 1$. The colored lines are a guide to the eye. Error bars represent the measurement standard error accounting for the linear fit to the $dR/dT$ data and assuming a base uncertainty of 1.2% in the resistance measurements. For the inset equation, $n_1$ and $n_2$ refer to the intra-flake carrier concentrations before and after an annealing step, respectively. See Supplementary Figure 5 for $dR/dT$ and $\eta$ analysis for all studied MXenes

Considering the role of intercalants first, we found that both $H_2O$ and $TBA^+$ intercalants increase sample resistance and can induce negative $dR/dT$. Vacuum annealing caused de-intercalation, significant decreases in sample resistance, and the onset of metallic conduction (except for $Ti_3CNT_x(TBA^+)$). The effects of $TBA^+$ intercalation on electronic properties were far greater than that of $H_2O$, and even after annealing at >750 °C, lingering effects of $TBA^+$ intercalation persisted. These findings are relevant for the optimization of MXene devices where large metallic conductivity is required and the increased inter-flake spacing associated with intercalation does not affect performance, e.g., wireless communication and wearable electronics. The ability to control $dR/dT$ through intercalation could find use in multi-functional sensors or in the development of MXene films with arbitrary $dR/dT$ values. Additionally, we note the striking similarities between the intercalated $Ti_3CNT_x$ and $Mo_2TiC_2T_x$ samples measured here with previous reports of semiconductor-like transport in Mo- and V-based MXenes[38–40]. Confirmation of intrinsic semiconducting MXene behavior will require temperature-dependent resistance measurements of single-flake MXene devices, which is beyond the scope of this study.

Regarding the effects of termination, vacuum annealing was shown to cause partial surface termination removal and increases in the MXene electronic conductivity. Oxygen terminations were more stable than –F terminations, and $Mo_2TiC_2T_x$ experienced a greater degree of surface de-functionalization than the Ti-based MXenes. These findings provide an avenue to further improve performance in MXene applications such as electromagnetic interference shielding and optoelectronics. For other applications, e.g., triboelectrics and catalysis, improved conductivity is desired, but the functionalized MXene surfaces offer chemical benefits. Hence, further analysis is needed to understand how partial de-

functionalization affects performance in these areas. From a broader perspective, our findings provide a first step towards realizing non-functionalized MXenes and termination-engineered MXenes, which are predicted to exhibit magnetism[56], fully spin-polarized transport[42], semiconducting behavior[36,38], and non-trivial topological order[60,78].

## Methods

**Syntheses of MXenes.** $Ti_3C_2T_x$ MXene was synthesized by selective etching of $Ti_3AlC_2$ MAX phase (Materials Research Center, Ukraine) in a mixture of LiF and HCl[6]. Specifically, 0.5 g of $Ti_3AlC_2$ powder was allowed to react with a premixed etchant (0.8 g of LiF and 10 mL of 9 M HCl) for 24 h at room temperature. Then the acid mixture was washed with 150 mL of deionized water for 3–5 cycles until the pH of the supernatant reached the value of 5. After that, the mixture was hand-shaken for 10 min followed by centrifugation at 3500 rpm for 10 min to remove unreacted MAX particles and reaction by-products. The dark supernatant was centrifuged at 3500 rpm for another 1 h to yield delaminated $Ti_3C_2T_x$ solution.

$Ti_3CNT_x$ MXene was synthesized from $Ti_3AlCN$ MAX[11] phase by two different routes. For the first route, the synthesis protocol is similar to the $Ti_3C_2T_x$ protocol mentioned earlier except that the reaction mixture was stirred at 500 rpm at 40 °C for 18 h. For the second route, etching is performed in HF acid and delamination is achieved with molecular intercalation. Initially, 2 g $Ti_3AlCN$ was etched in 20 mL of ~30% HF acid for 24 h at room temperature. The reaction mixture was washed with 150 mL of deionized water for 3–5 cycles until the pH of the supernatant reached the value of 5. The resulting mixture was filtered through a filter paper (3501 Coated PP, Celgard, USA) to collect multilayer $Ti_3CNT_x$ powder. To delaminate the $Ti_3CNT_x$ powder, 1 g of dry powder was stirred in a mixture of 9 mL of water and 1 mL of tetrabutylammonium hydroxide (TBAOH) solution (48 wt%, Sigma Aldrich) for 24 h. The mixture was hand-shaken for 10 min and washed with 3–5 cycles of deionized water. After neutral pH was reached, the mixture was centrifuged at 3500 rpm for 10 min to remove unreacted MAX particles and reaction by-products. The brownish supernatant was centrifuged at 3500 rpm for another hour to yield delaminated $Ti_3CNT_x$ solution.

$Mo_2TiC_2T_x$ was synthesized from $Mo_2TiAlC_2$ MAX phase[59]. To produce the MAX phase, Mo, Ti, Al, and graphite powders (Alfa Aesar, Ward Hill, MA) were mixed in a ratio 2:1:1.1:2 and ball milled for 18 h in a plastic jar with zirconia balls. The powder mixture was transferred in an alumina crucible and held at 1600 °C for 4 h (5 °C min$^{-1}$, Ar). The resulting MAX block was drilled and sieved (400 mesh, particle size <38 μm). Specifically, 2 g of $Mo_2TiAlC_2$ powder was added to 40 mL of 48–51% aqueous HF solution and stirred at 50 °C for 48 h. The mixture was washed and collected the same way as the HF-prepared $Ti_3CNT_x$ MXene. To delaminate $Mo_2TiC_2T_x$ powder, 1 g of the powder was stirred in 10 mL of 48% TBAOH aqueous solution for 12 h. The mixture was washed with 3 cycles of deionized water (3500 rpm for 15 min, each cycle). After a pH of 7–8 was reached, the supernatant was decanted the same way as the $Ti_3CNT_x(TBA^+)$. The sediment was dispersed in 30 mL of deionized water and sonicated for 30 min in a water-ice sonication bath under argon bubbling. The final solution was centrifugate at 3500 rpm for 30 min. Finally, the supernatant is collected.

We note that the exact changes in conductivity, intercalation, and surface termination reported here are likely related to MXene film thickness, flake size, and general flake quality, which are all a function of the synthesis process. As such, the measurement of different MXene films—produced with differing synthesis procedures—will likely produce a differing degree of conductivity changes with annealing.

**Electron microscopy and spectroscopy.** A JEOL 2100F microscope with a Schottky electron emitter was used for these experiments. The microscope is equipped with the high-resolution pole piece with a $C_S$ of 1 mm. EELS measurements were performed in TEM (diffraction) mode in order to sample a large area of the MXene film and to reduce irradiation-induced sample degradation. The EELS collection angle was set to 11 mrad. A direct detection (DD) EELS system was used as recently reported in ref. [53]. The energy dispersion was set to 0.125 eV per channel, and the zero-loss peak (approximate energy resolution) was measured to be ~1.0 eV. For EELS analysis, Gatan DigitalMicrograph was used.

**In situ heating and biasing.** The DENSsolutions Lighting D9+ heating and biasing sample holder was used for in situ TEM/EELS experiments with the 8 contact (A1-type) heating and biasing nanochips[62]. MXenes were deposited onto the nanochips through spray casting[24] (Supplementary Figure 2). A mask was used to confine the MXene deposition to an area centered on the biasing electrodes (Supplementary Figure 2). Resistance measurements were performed with a 4-probe geometry using a Keithley 2400 SMU. Due to the high conductivity of MXene and the large sample cross-sectional area (relative to in situ TEM standards), application of voltages above 0.1 V led to high current densities. The resulting Joule heating strongly affected heating measurements and damaged the heating and biasing chips. As such, the resistance measurements were obtained with applied voltages of ≤5 mV.

The maximum annealing temperatures were 775, 700, 750, and 775 °C, for $Ti_3C_2T_x$, $Ti_3CNT_x$, $Ti_3CNT_x(TBA^+)$, and $Mo_2TiC_2T_x$, respectively. The $Ti_3CNT_x$ samples were not heated to 775 °C since previous experiments showed a larger extent of $TiO_2$ formation at 775 °C. Heating and cooling rates were 1 °C s$^{-1}$. For annealing steps performed at 75, 125, and 200 °C, the maximum temperature was held for 10 s, and for annealing steps performed at 300, 500, 700, and 775 °C, the maximum temperature was held for 5 min. We note that the average temperature within the samples deviated from the input temperature due to temperature gradients within the films (Supplementary Figure 10).

A different heating procedure was used for the $Ti_3CNT_x(TBA^+)$ sample. Initially, free $TBA^+$ molecules prevented good electrical contact, and the sample was annealed in situ at 200 °C to achieve adequate contact between the MXene film and Pt electrodes. Then the sample was heated to 750 °C and held at this temperature for 1 h. This modified procedure was aimed at ensuring maximum de-intercalation/decomposition of the $TBA^+$ molecules without significant $TiO_2$ formation.

We note that for each $M_{n+1}X_n$ chemistry, two samples were tested with in situ TEM heating and biasing, and each sample showed qualitatively similar behavior (Supplementary Table 2).

**Thermogravimetric-mass spectrometry analysis.** Simultaneous thermogravimetric-mass spectrometry analysis was performed on a Discovery SDT 650 model connected to Discovery mass spectrometer (TA Instruments, DE). Vacuum-filtered films of MXenes were cut to small pieces and packed in a 90 μL alumina pan. Before heating, the analysis chamber was flushed with He gas at 100 mL min$^{-1}$ for 1 h to reduce residual air. The samples were heated to 1500 °C at a constant heating rate of 10 °C min$^{-1}$ in He atmosphere (100 mL min$^{-1}$).

**Low temperature electronic transport.** Electronic transport properties of MXenes after heat treatments were measured in a Quantum Design EverCool II Physical Property Measurement System (PPMS). After the in situ annealing experiments, the heating and biasing nanochip was removed from the TEM and wired to the PPMS sample puck using conductive Ag paint (Supplementary Figure 11). Temperature-dependent resistance was recorded from 25 °C (300 K) down to −263 °C (10 K) in a low pressure helium environment (~20 Torr). Magnetoresistance data was recorded at 10 K by sweeping the magnetic field-perpendicular to the chip from −70 kOe to 70 kOe. We note that for both the $Ti_3CNT_x$ and $Mo_2TiC_2T_x$ samples, the room temperature resistance and d$R$/d$T$ values (but not the sign of d$R$/d$T$) differed between the final measurement within the TEM and subsequent measurements within the PPMS. This difference is likely due to the differing experimental set-up and/or sample degradation during exposure to atmosphere after removal from the TEM.

**X-ray photoelectron spectroscopy.** X-ray photoelectron spectroscopy analysis was performed in a spectrometer (Physical Electronics, VersaProbe 5000, Chanhassen, MN) using a 100 μm monochromatic Al Kα X-ray beam. Photoelectrons were collected by a takeoff angle of 45° between the sample surface and the hemispherical electron energy analyzer. Charge neutralization was applied using a dual beam charge neutralizer irradiating low-energy electrons and ion beam. Vacuum-filtered films were mounted on double-sided tape and were electrically grounded using a copper wire. Prior to data acquisition, an Ar beam operating at 2 kV and 2 μA was used to sputter the sample surface for 2 min. Quantification and deconvolution of the core-level spectra was performed using a software package (CasaXPS Version 2.3.16 RP 1.6). Background contributions to the measured intensities were subtracted using a Shirley function prior to quantification and deconvolution.

**X-ray diffraction.** The X-ray diffraction data were acquired from vacuum-filtered films of the same solution used for the in situ TEM study. The data were acquired by a diffractometer (Rigaku Smart Lab, USA) with Cu Kα radiation at a step size of 0.03° with 0.6 s dwelling time.

## Data availability

The datasets generated and/or analyzed during the current study are available from the corresponding author on reasonable request.

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

## Acknowledgements

J.L.H., A.C.L., and M.L.T. acknowledge funding from the National Science Foundation Major Research Instrumentation award #DMR-1429661, supporting electron microscopy, including EELS, and high-temperature transport measurements. K.H., Y.G., B.A., D.P., and S.M. acknowledge funding from the U.S. Department of Energy (DOE), Office of Science, Office of Basic Energy Sciences, grant #DE-SC0018618, supporting MXene synthesis, low-temperature transport, photoemission, and thermogravimetric analysis. J. L.H., A.C.L., and M.L.T. thank Ben Miller, Stephen Mick, and Paolo Longo of Gatan Inc. for helpful discussions regarding EELS acquisition and in situ heating and biasing measurements. Tyler Mathis, Simge Uzun, and Mohamed Alhabeb are acknowledged for providing MXene solutions. Drexel University Core Facilities is acknowledged for providing access to XPS, XRD, and TEM instruments. Acquisition of the PPMS was supported by the U.S. Army Research Office under grant No. W911NF-11-1-0283.

## Author contributions

M.L.T. and Y.G. conceived of the experimental plan. J.L.H. performed the in situ TEM and EELS experiments with assistance from A.C.L., K.H., and Y.P. K.H. synthesized and deposited MXenes samples, performed TGA-MS, PPMS, and XPS measurements. B.A. and D.P. prepared $Mo_2TiAlC_2$ and $Mo_2TiC_2T_x$, respectively. Y.P. and J.T.v.O. performed nanochip temperature simulations. J.L.H. prepared the manuscript. All authors reviewed and contributed to the final manuscript.

## Additional information

**Competing interests:** Y.P. and J.T.v.O. are employees of DENSsolutions, which developed and is marketing the Lightning D9+ sample holder used here. The remaining authors declare no competing interests.

