## [Peer Review File · Nature Communications]

Reviewers' comments:

Reviewer #1 (Remarks to the Author):

The manuscript summarizes results on the de-intercalation and removal of specific terminations on the well-known emerging class of 2D materials, i.e. MXenes. The manuscript covers three types of MXenes. The work is based on previous results by the same group of authors and others in the literature. The report provides a nice qualitative report on changes of electronic properties after thermal treatment. Experiments are performed in the low vacuum atmosphere of a transmission electron microscope. The observed changes of the conductivity and the possibility of control of semiconducting versus metallic behavior may become publishable, if the following problems are resolved.

1) Experimentally, the interface contact(s) in a 2 point measurement of electric transport must be known. This does not seem to be the case here, and the validity of the data may be questioned. I believe that it is necessary that the authors provide a four point measurement of one single sheet (for analysis of the effect of termination) and one double/triple layer complemented for example by XPS or Auger spectroscopy to validate the interpretation of their measurements.

2) One can find quite a few publications in the literature which discuss the effect of changes in termination for example in studies of electrocatalytic hydrogen evolution. In such studies the effect of catalytic changes is referred to changes of the conductivity due to termination. A connection should be drawn.

I cannot recommend acceptance.

Reviewer #2 (Remarks to the Author):

Hart et al provide a report detailing in situ transmission electron microscope (TEM) resistance measurements for spray deposited multflake MXene films – Ti₃C₂, Ti₃CN and Mo₂TiC₂. They measure resistance changes while sequential cyclic annealing of the samples to higher temperatures (up to 775C). They use post- annealing electron energy loss spectroscopy to analyse the O-K edge and F-K edges for the samples to provide understanding of the changes to intercalant molecules and surface terminating groups as a function of annealing temperature. They complement the TEM results with thermal gravimetric analysis-mass spectrometry and subsequent measurements of the temperature dependent resistance.

Unfortunately I am not convinced there is great novelty in the results. For example, it is not surprising that higher annealing temperatures are required to remove CTAB vs water and oxygen, or that the latter begin to be removed as soon as the sample is introduced to the vacuum. Figure S2 shows that very large time-dependent resistance changes occur at room temperature when the sample enters the vacuum. However the potential for time dependent changes in the measurements presented in Fig 1 is missing.

The uncontrolled structure of the films being tested is also likely to have a significant influence on the results (as flake thicknesses, film thickness and flake sizes are likely to play an important role in determining the deabsorption behaviour). Furthermore, the lack of error bars or repeat measurements for the TEM data is also a concern. The use of in situ TEM has the main advantage of provides access to high spatial resolution microstructural data but this is not used for this work. Although I accept it is important to minimise the potential for beam induced damage of the materials, the lack of high spatial resolution means that the resistance experiments could have been performed more reliably on bulk films using traditional device fabrication methods. The lack of complementary ex-situ characterisation of resistivity changes for these materials is a significant weakness of the manuscript.

In addition I would like to highlight the following formatting/typographic suggestions:

I found the structure of the manuscript slightly hard to follow – for example the introduction seems to assume significant knowledge of the synthesis of MXenes from MAX phase and that these materials exist as thin (few/many? layer flakes). This is not stated until the results. The summary of the intercalation section reads like a conclusion but is before the discussion of the termination results.

Suppl Fig 5 shows Ti₃CNT_x data but refers to TiC₂T_x data (not displayed?)

Line 324: Perrson should be Persson?

Line 375 Semicondcutor should be semiconductor

Line 251, 253, 446: 'lead' should be 'led'

The references are strongly dominated by self citations.

Re: Control of MXenes' Electronic Properties Through Termination and Intercalation

We thank the reviewers for their helpful comments, and in response, we have considerably updated the manuscript and the Supplementary Information with new data and analysis. All new data in the manuscript and Supplementary is marked in red. In this document, we respond to the reviewer comments point by point. The original reviewer comments are indented and shown in red text, and our responses are in black. When we refer to figures and line numbers in the manuscript and Supplementary Information, it is with respect to the updated manuscript and Supplementary Information. Figures shown within this document are referred to as Fig. R1, Fig. R2, etc.

Reviewer #1 (Remarks to the Author):

The manuscript summarizes results on the de-intercalation and removal of specific terminations on the well-known emerging class of 2D materials, i.e. MXenes. The manuscript covers three types of MXenes. The work is based on previous results by the same group of authors and others in the literature. The report provides a nice qualitative report on changes of electronic properties after thermal treatment. Experiments are performed in the low vacuum atmospheres of a transmission electron microscope. The observed changes of the conductivity and the possibility of control of semiconducting versus metallic behavior may become publishable, if the following problems are resolved.

1) Experimentally, the interface contact(s) in a 2 point measurement of electric transport must be known. This does not seem to be the case here, and the validity of the data may be questioned.

All reported resistance measurements were conducted with a 4-point geometry. This is stated in the Methods section (line 542) and shown in Supplementary Figure 2.

I believe that it is necessary that the authors provide a four point measurement of one single sheet (for analysis of the effect of termination) and one double/triple layer complemented for example by XPS or Auger spectroscopy to validate the interpretation of their measurements.

We agree with the reviewer that single-, double-, and multi-flake electrical and spectroscopic measurements would be a comprehensive tool to verify the effects of intercalation and surface terminations on MXenes' electronic properties. However, such measurements are difficult to accomplish due to fairly small lateral size of MXene flakes produced and the nature of wet chemical synthesis of MXene etching. We are actively working on developing synthesis and measurement methods to accomplish such measurements, which we hope to report separately from this present work.

However, we believe that the results shown in this study are adequate to support our conclusions. The main findings of our work are 1) at low annealing temperatures de-intercalation can cause changes in the sign of dR/dT due to inter-flake effects, and 2) at higher annealing temperatures the loss of termination species increases the MXene intra-flake conductivity through an increase in carrier concentration. Below, we argue that the data included in the original manuscript fully supports both of these claims.

Addressing point number (2) above, we believe that the data included in the original submission proves termination control over electronic conductivity. Specifically, after annealing Ti_3CNT_x and $Ti_3C_2T_x$ past 500 °C, we show that there is no further H_2O loss, confirmed with both *in situ* EELS of the O *K*-edge (Fig. 2b and 3c) and *ex situ* TGA-MS of H_2O release (Fig. 2a and Fig. 3b). Additionally, previous XRD reports have shown that while there is a significant change in the *c*-lattice spacing of $Ti_3C_2T_x$ after annealing in Ar at 500 °C, there is no additional lattice change when annealing at temperatures above 500 °C¹. Since there is no loss of intercalation and

no change in the inter-flake spacing when annealing these MXenes above 500 °C, we conclude that annealing Ti_3CNT_x and $\text{Ti}_3\text{C}_2\text{T}_x$ above 500 °C does not alter their inter-flake resistance. However, our experiments show that after annealing at 500 °C, the resistances of $\text{Ti}_3\text{C}_2\text{T}_x$ and Ti_3CNT_x continue to decrease during annealing steps at 700 and 775 °C. If this change in resistance is not due to a change in the MXene inter-flake resistance, it must be due to a change in the intra-flake resistance. Indeed, at these temperatures we observe the partial loss of -F with *in situ* EELS (Fig. 3a and Supplementary Figure 8) and *ex situ* TGA-MS (Fig. 3b and Supplementary Figure 4). Importantly, the *only* chemical or structural change observed in these samples with *in situ* EELS, *in situ* TEM imaging, *in situ* electron diffraction, and *ex situ* TGA-MS when annealing from 500 to 775 °C is the loss of -F termination species (and a small degree of TiO_2 formation which cannot account for the increased conductivity). This is conclusive evidence that the decrease in sample resistance with annealing beyond ~500 °C for these MXenes is due to the loss of termination species.

To further prove this point, we analyzed the relative changes in MXene dR/dT with annealing. Because $\text{Ti}_3\text{C}_2\text{T}_x$ displays metallic behavior at all annealing temperatures, we assume its transport can be described with the Drude metal formula. Within the Drude theory of conductivity, a metal's resistance is given by

$$R = \frac{m\omega}{e^2n} \quad \text{Equation 1}$$

where m is the effective electron mass, ω is the electron scattering rate, e is the charge, and n is the carrier concentration. For the temperature range of our *in situ* TEM measurements (RT up to 775 °C), electron scattering is due to impurity scattering and phonon scattering. While impurity scattering is temperature independent, electron-phonon scattering is linear in temperature (at RT and above)^{2,3}. As such

$$\frac{dR}{dT} \propto \frac{m}{e^2n} \quad \text{Equation 2}$$

Hence, both R and dR/dT are inversely proportional to the carrier concentration, n . During our *in situ* annealing measurements, if the only decrease in MXene resistance is due to an increase in the intra-flake carrier concentration, then the proportional decrease in dR/dT should be equal to the proportional change in resistance. This relation holds true for $\text{Ti}_3\text{C}_2\text{T}_x$ when annealing at 500, 700, and 775 °C, and it is accurate for annealing Ti_3CNT_x at 700 °C (see Fig. 4 and Supplementary Information discussion for a more in depth explanation). We reiterate that for these annealing steps, the relative changes in dR/dT closely match what one would expect if the change in resistance were due to a change in carrier concentration. DFT has predicted that changing the MXene termination alters the carrier concentration⁴. Thus, our analysis of changes in dR/dT provide further proof that when we vacuum anneal $\text{Ti}_3\text{C}_2\text{T}_x$ and Ti_3CNT_x above 500 °C, the measured changes in resistance are due to termination loss and an associated increase in the intra-flake carrier concentration.

We next discuss our claim that intercalation can drive negative dR/dT in MXenes. First, we consider $\text{Ti}_3\text{C}_2\text{T}_x$. In its as-prepared state, this MXene displays metallic transport, thus the intra-flake conduction must be metallic. When this MXene is annealed at 125 and 200 °C, the resistance decreases, but the value of dR/dT *increases* (Supplementary Table 2). Within the

Drude theory of metals (Equations 1 and 2 in this document), there is no single mechanism, e.g. a change in impurity scattering, effective carrier mass, electron-phonon scattering, or charge carrier density, which can simultaneously decrease a metal's resistance and increase the value of dR/dT . Thus there must be an additional term contributing to the macroscopic MXene resistance which is responsible for the simultaneous decrease in resistance and increase in dR/dT , i.e. there must be a contribution from the inter-flake resistance. If the inter-flake resistance was metallic in nature, then, similar to the intra-flake metallic term, the increase in dR/dT with annealing at 125 and 200 °C could not be explained. However, an insulating inter-flake resistance can account for the increase in dR/dT given a decrease in resistance. For the insulating models used to describe semiconductor-like MXenes, i.e. activated transport and variable range hopping⁵, any decrease in resistance must be accompanied by an increase in dR/dT (see Supplementary Information for a mathematical derivation of this argument). To summarize this argument, the simultaneous decrease in resistance and increase in dR/dT of $Ti_3C_2T_x$ during annealing at 125 and 200 °C is not explained by any change to the intra-flake properties, but it is consistent with a decrease in the inter-flake resistance (if the inter-flake term is insulating). In accord with this resistance and dR/dT analysis, we measured H_2O de-intercalation from $Ti_3C_2T_x$ during annealing at these low temperatures with both *in situ* EELS (Fig. 3c) and *ex situ* TGA-MS (Fig. 3b), and, it is known that annealing $Ti_3C_2T_x$ at these temperatures reduces the inter-flake spacing⁶ (due to de-intercalation). Taken together, this data conclusively proves that the inter-flake resistance in $Ti_3C_2T_x$ is insulating in nature, and that it is affected during low temperature annealing due to de-intercalation.

Having demonstrated that the inter-flake resistance in $Ti_3C_2T_x$ is insulating, it is logical that the inter-flake resistance in Ti_3CNT_x and $Mo_2TiC_2T_x$ is also insulating in nature. Due to the increased inter-flake separation (larger *c*-lattice parameter) in these MXenes compared to $Ti_3C_2T_x$ (Fig. 2c and ref. 5), it is expected that the relative strength of the insulating inter-flake resistance increases in these MXenes with respect to $Ti_3C_2T_x$. This relative increase in the (insulating) inter-flake resistance then causes the transition from positive to negative dR/dT for the macroscopic MXene behavior. Supporting these arguments, we observe a transition from negative to positive dR/dT in Ti_3CNT_x when annealing at 200 °C. At this temperature, there is no evidence of $-F$, $-OH$, or $-O$ termination loss (Fig. 2a,b and Supplementary Figures 4 and 8), and more broadly, there is no experimental data which suggests a change in the intra-flake resistance. If there is no change to the intra-flake resistance, then the changes in resistance and dR/dT must be due to a change in the inter-flake resistance. Indeed, we observe de-intercalation of H_2O at these temperatures with *ex situ* TGA-MS (Fig. 2a) and *in situ* EELS (Fig. 2b), and, as a result of de-intercalation, we measure a decrease in the inter-flake spacing with *ex situ* XRD (Fig. 2c). In total, this data demonstrates conclusively that the transition from negative to positive dR/dT in Ti_3CNT_x is due to de-intercalation of H_2O .

To make the above points clearer in the manuscript, we have moved several figures from the Supplemental Information into the manuscript, and we have added in additional analysis and citations to the main text. Specifically, we have added *ex situ* TGA-MS and *in situ* EELS of $Ti_3C_2T_x$ to the manuscript (Fig. 3b,c). This data shows that in $Ti_3C_2T_x$, de-intercalation of H_2O is complete by ~400 °C, indicating that changes in resistance for annealing above ~400 °C cannot be due to de-intercalation. Additionally, the added TGA-MS data shows the loss of $-F$ termination species beginning at ~400 °C. We have also cited ref. 1 from this document in the

manuscript (it is ref. 74 in the manuscript), which shows that there is no change in the $Ti_3C_2T_x$ lattice parameter with annealing beyond 500 °C. We have also added the *in situ* EELS of the O *K*-edge of Ti_3CNT_x to the manuscript, which shows that H_2O de-intercalation occurs during *in situ* annealing steps at 125 and 200 °C, concurrent with the onset of metallic conduction (Fig. 2b). We have added *ex situ* XRD of Ti_3CNT_x , showing that annealing at 150 °C results in a smaller *c*-lattice parameter (Fig. 2c). We have added *in situ* time-resolved EELS data to Figure 3e,f, which shows time-resolved, simultaneous decreases in $-F$ termination concentration (measured with EELS) and increased conductivity. This time-resolved data provides an additional connection between termination loss and decreased resistance. Lastly, we have added a new section to the Supplementary Information, ‘MXene dR/dT analysis’, which expands upon the above analysis relating to the Drude formula and insulating inter-flake formula. This dR/dT analysis is briefly explained in the manuscript text, and summarized in Fig. 4. We have rewritten a large extent of the manuscript text (shown in red) to make the preceding arguments more explicit.

2) One can find quite a few publications in the literature which discuss the effect of changes in termination for example in studies of electrocatalytic hydrogen evolution. In such studies the effect of catalytic changes is referred to changes of the conductivity due to termination. A connection should be drawn.

We thank the reviewer for their suggestion. We have added electrocatalysis to the list of MXene applications which are influenced by the electronic conductivity and to the list of MXene functional properties which are predicted to be influenced by termination.

Reviewer #2 (Remarks to the Author):

Hart et al provide a report detailing *in situ* transmission electron microscope (TEM) resistance measurements for spray deposited multflake MXene films – Ti_3C_2 , Ti_3CN and Mo_2TiC_2 . They measure resistance changes while sequential cyclic annealing of the samples to higher temperatures (up to 775C). They use post-annealing electron energy loss spectroscopy to analyse the O-K edge and F-K edges for the samples to provide understanding of the changes to intercalant molecules and surface terminating groups as a function of annealing temperature. They complement the TEM results with thermal gravimetric analysis-mass spectrometry and subsequent measurements of the temperature dependent resistance.

Unfortunately I am not convinced there is great novelty in the results. For example, it is not surprising that higher annealing temperatures are required to remove CTAB vs water and oxygen, or that the latter begin to be removed as soon as the sample is introduced to the vacuum.

We agree with the reviewer that the relative temperatures of TBA⁺, H₂O, and adsorbed O₂ loss are not surprising or particularly novel. The novelty of our work is relating these processes with changes in the MXene electronic properties which are crucial to the MXene performance. This is very critical fundamental knowledge about MXenes which is currently lacking in the literature.

Despite myriad theoretical predictions of termination loss effecting electronic resistance, we provide the first direct experimental measurements of termination effects on MXene electronic properties. This is an important finding in several regards: advancing our fundamental understanding of MXene electronic properties; optimizing MXene devices which depend on conductivity; and providing a first step towards realizing termination-engineered MXenes for semiconductor and spintronic applications.

Beyond the novelty of correlating termination loss and improved conductivity, we are the first to show that intercalation can control the sign of dR/dT . This finding is critical for the community's understanding of previously published semiconductor-like MXenes. Additionally, the demonstrated control of dR/dT *via* intercalation offers new functionality to the MXene family and could find use in novel sensors or devices which require specific values of dR/dT .

Simply put, our work provides the most direct and in depth experimental study of the effects of intercalation and termination on MXene electronic conductivity. Given the state-of-the-art performance MXenes show for applications, e.g. supercapacitors, and the dependence of this performance on electronic properties, our work is crucial to the further development of MXene devices.

Figure S2 shows that very large time-dependent resistance changes occur at room temperature when the sample enters the vacuum. However the potential for time dependent changes in the measurements presented in Fig 1 is missing.

Below we show the time-resolved resistance changes from the data presented in Fig. 1. Additionally, we show coupled time-resolved resistance measurements and time-resolved EELS for a Ti₃CNT_x sample. All of this data has been added to the Supplementary Information, and for the time-resolved EELS, it has been added to the main text.

Fig. R1. Time resolved resistance measurements of $\text{Ti}_3\text{C}_2\text{T}_x$ during various *in situ* annealing steps. The data is the same as that presented within Fig. 1 of the text. Heating and cooling rates were $1\text{ }^\circ\text{C/s}$. In between heating and cooling, the sample was held for 10 s for annealing temperatures of 75 to 200 $^\circ\text{C}$, and 5 min for 300 to 775 $^\circ\text{C}$. In addition to labeling the initial temperature (25 $^\circ\text{C}$) and annealing temperature, we also mark the temperature where the previous annealing step was performed (marked with a star). In some cases, not all tick marks on the top x-axis (the temperature axis) could be labeled. The unmarked ticks in **b**, **c**, and **g**, correspond to 125, 200, and 775 $^\circ\text{C}$, respectively.

Fig. R2. Time resolved resistance measurements of Ti_3CNT_x during various *in situ* annealing steps. The data is the same as that presented within Fig. 1 of the text. Heating and cooling rates were $1^{\circ}C/s$. In between heating and cooling, the sample was held for 10 s for annealing temperatures of 75 to 200 $^{\circ}C$, and 5 min for 300 to 700 $^{\circ}C$. In addition to labeling the initial temperature (25 $^{\circ}C$) and annealing temperature, we also mark the temperature where the previous annealing step was performed (marked with a star). In some cases, not all tick marks on the top x-axis (the temperature axis) could be labeled. The unmarked ticks in **b** and **c** correspond to 125 and 200 $^{\circ}C$, respectively.

Fig. R3. Time resolved resistance measurements of $\text{Mo}_2\text{TiC}_2\text{T}_x$ during various *in situ* annealing steps. The data is the same as that presented within Fig. 1 of the text. Heating and cooling rates were 1 °C/s. In between heating and cooling, the sample was held for 10 s for annealing temperatures of 75 to 200 °C, and 5 min for 300 to 775 °C. In addition to labeling the initial temperature (25 °C) and annealing temperature, we also mark the temperature where the previous annealing step was performed (marked with a star). In some cases, not all tick marks on the top x-axis (the temperature axis) could be labeled. The unmarked ticks in **b**, **c**, and **g**, correspond to 125, 200, and 775 °C, respectively.

Fig. R4. **a**, Change in resistance of Ti_3CNT_x with *in situ* annealing. We used a different annealing sequence on this sample compared to the Ti_3CNT_x sample shown in the manuscript Fig. 1b. The first annealing step for this sample was at 300 $^{\circ}C$, and during this annealing step, there was a transition in dR/dT . **b**, **d**, and **f**, show the time-resolved changes in resistance for each annealing step. For comparison, **c**, **e**, and **g** show the time-resolved normalized intensity of the fluorine *K*-edge, measured with *in situ* EELS. Each individual spectrum, shown with black dots, had an exposure time of 4 seconds. The green line is a smooth of the data. The stars in **b-g** show the maximum temperature of the prior annealing step.

Figures R1, R2, and R3 show the time-resolved resistance changes for $Ti_3C_2T_x$, Ti_3CNT_x , and $Mo_2TiC_2T_x$, respectively. The stars represent the temperature at which the sample surpasses the maximum previous annealing temperature. Generally, after surpassing this temperature, the samples displayed non-uniform dR/dT owing to intercalation and termination loss.

For all of the samples shown in the original manuscript, EELS was only acquired pre- and post-annealing. However, additional samples were studied with time-resolved EELS during

annealing. Following the reviewer's comment, we have added the time-resolved electronic resistance and EELS data for a Ti_3CNT_x (LiF and HCl etched) sample into the manuscript (Fig. R4). This sample was annealed at 300 °C, 500 °C, and 700 °C. During the annealing steps at 500 and 700 °C, the time-resolved F *K*-edge intensity shows the loss of –F termination species as the sample resistance is decreasing. This time-resolved simultaneous measurement of –F termination loss and reduced electronic resistance provides further evidence of termination control over conductivity.

Figures R1-R3 have been added to the Supplementary Information (Supplementary Figures 13-15) and are referred to in the Fig. 1 caption. Figure R4 has also been added to the Supplementary Information (Supplementary Figure 9) and some of this data has been added to the main text (Fig. 3e,f).

The uncontrolled structure of the films being tested is also likely to have a significant influence on the results (as flake thicknesses, film thickness and flake sizes are likely to play an important role in determining the deabsorption behaviour).

We agree that changes in film thickness, flake size, and general flake quality will alter the sample electronic resistance and the desorption behavior. In fact, we tested multiple samples for each MXene chemistry (this is discussed more in the next section), and while the different samples showed qualitatively similar behavior, there were differences in their behavior. For example, the $\text{Mo}_2\text{TiC}_2\text{T}_x$ sample presented in the manuscript showed a 6 times increase in conductivity with annealing, while a secondary $\text{Mo}_2\text{TiC}_2\text{T}_x$ sample showed an 8 times increase in conductivity with annealing. These differences are attributed to the aspects of the MXene samples which are not controlled, e.g. flake size. However, we stress that the main results of this study, namely, 1) that intercalation can control dR/dT and 2) that termination loss increases electronic conductivity, did not vary between the studied samples (see next section). Thus, we acknowledge that there are uncontrolled aspects of the studied samples (e.g., film thickness) and this prevents quantification of some properties (e.g., absolute value of the film conductivity) and may influence desorption kinetics; however, the main findings of this paper are not affected by the uncontrolled aspects of our samples.

The following text has been added to the methods section, line 525, “We note that the specific changes in conductivity, de-intercalation, and de-functionalization reported here are likely related to MXene film thickness, flake size, and general flake quality, which are all a function of the synthesis process. As such, the measurement of different MXene films – produced with differing synthesis procedures – will likely produce a differing degree of conductivity changes with annealing.”

Furthermore, the lack of error bars or repeat measurements for the TEM data is also a concern.

We repeated the *in situ* annealing and biasing inside the TEM at least 2 times for each MXene sample, though this was not mentioned in the original manuscript. We have added a table to the Supplementary Information (Supplementary Table 1) which summarizes our measurements of

MXene samples not included within the manuscript. We have added the following text to the paper (line 562) “We note that for each $M_{n+1}X_n$ chemistry, two samples were tested with *in situ* TEM heating and biasing, and each sample showed qualitatively similar behavior (Supplementary Table 1).” Supplementary Table 1 is shown below.

Supplementary Table 1 | Summary of *in situ* heating and biasing results not included in the manuscript. The F and O columns give the intensity of fluorine and oxygen *K*-edges after the final annealing step, normalized to the initial edge intensity, *i.e.* a value of 1 means no change in intensity, and a value of 0 means the complete disappearance of the edge. R_i/R_A gives the ratio of the initial to the final resistance. For these samples, the resistance was not measured prior to insertion into the TEM, so changes in resistance do not reflect the desorption of adsorbed molecules. The initial resistance was measured within the TEM. The $Ti_3CNT_x(TBA^+)$ sample was too thick to obtain meaningful EELS data.

MXene Chemistry	As-prepared		Max Annealing T °C	Annealed				
	Intercalant	dR/dT		F	O	Intercalant	dR/dT	R_i/R_A
$Ti_3C_2T_x$	H_2O, Li^+	+	790	0.5	0.9	Li^+	+	1.5
Ti_3CNT_x	H_2O, Li^+	–	700	0.5	0.9	Li^+	+	4.5
$Ti_3CNT_x(TBA^+)$	H_2O, TBA^+	–	750	-	-	-	–	>10
$Mo_2TiC_2T_x$	H_2O, TBA^+	–	650	0	0.6	-	+	8

The use of *in situ* TEM has the main advantage of provides access to high spatial resolution microstructural data but this is not used for this work. Although I accept it is important to minimise the potential for beam induced damage of the materials, the lack of high spatial resolution means that the resistance experiments could have been performed more reliably on bulk films using traditional device fabrication methods.

Even in the absence of high resolution imaging, we argue that our *in situ* TEM set-up offers a unique platform to study the effects of intercalation and termination on MXenes’ electronic properties. To directly correlate MXene termination loss with changes in electronic resistance, several experimental requirements should be met: 1) the sample must be heated in an inert environment (heating in atmosphere would cause rapid oxidation) up to at least 500 °C to eliminate intercalation species and induce the loss of termination species, 2) electronic resistance measurements must be performed during annealing or prior to the exposure of the sample to vacuum, since atmospheric exposure causes surface re-functionalization and changes to the sample conductivity, 3) *in situ* chemical analysis to measure changes in the sample intercalation and termination, and 4) *in situ* structural analysis and imaging to detect any phase transitions, *i.e.* $Ti_3C_2O_x \rightarrow TiO_2$ or the formation of voids⁷. Our *in situ* TEM set-up meets all of these requirements, and we are not aware of any other experimental technique which meets all of these requirements. *In situ* XAS or XPS could allow *in situ* heating and biasing and chemical analysis, but these techniques do not allow structural characterization.

Secondly, we stress that high resolution experiments were not practical given the MXene electron beam sensitivity. Recent work shows that $Ti_3C_2T_x$ loses –O termination species and undergoes significant structural changes when subjected to high resolution STEM imaging at 500 °C⁷ (see Fig. R5a,b). Clearly, the structural break down of the Ti_3C_2 shown in ref. 7 would prevent any correlation between MXene termination and conductivity. Additionally, electron beam induced loss of termination species would constitute a significant problem. If the termination loss we measure with EELS is due to the electron beam, we cannot correlate changes

in termination and electronic resistance, since our resistance measurements are across a MXene area far greater than the area exposed to the electron beam. Thus, to accurately correlate changes in MXene electronic resistance and changes in intercalation and termination, it is critical to keep the electron dose (and current density) low enough to avoid destruction of the Ti_3C_2 structure and beam-induced loss of termination species. Figure R5c shows the $\text{Ti}_3\text{C}_2\text{T}_x$ sample we investigated after annealing at $775\text{ }^\circ\text{C}$. While TiO_2 is seen in our sample, this is a natural result of annealing Ti_3C_2 , i.e. it is not beam induced. Importantly, we do not observe voids in our film. Moreover, we do not see the disappearance of the O *K*-edge (see manuscript Fig. 2b and 3c), indicating that we are not removing termination species with the beam. Thus, by spreading the TEM beam and reducing the electron intensity, we are able to avoid beam induced damage, albeit at the necessary expense of high resolution imaging. We have added additional TEM imaging data (Supplementary Figure 1) to demonstrate that under our low-dose imaging and spectroscopy conditions, we do not have significant beam-induced sample damage.

Fig. R5. **a**, STEM image taken from ref. 7 showing the effect of high resolution imaging at $500\text{ }^\circ\text{C}$ on the structure of $\text{Ti}_3\text{C}_2\text{T}_x$ MXene. In addition to structural break down, intense electron irradiation can induce the loss of $-\text{O}$ termination species (**b**). In our work, we keep the electron beam intensity extremely low to avoid beam induced loss of termination species and to maintain the structural integrity of our MXene flakes. For instance, **c** shows $\text{Ti}_3\text{C}_2\text{T}_x$ after annealing to $775\text{ }^\circ\text{C}$.

The lack of complementary *ex-situ* characterisation of resistivity changes for these materials is a significant weakness of the manuscript.

Below, we explain that for all of the samples shown in Fig. 1, *ex situ* electrical characterization was performed both pre- and post- *in situ* annealing. Additionally, we present *ex situ* measurements of bulk films which reproduce the transition in dR/dT for Ti_3CNT_x and $\text{Mo}_2\text{TiC}_2\text{T}_x$.

All of the samples shown in Fig. 1 were biased in atmosphere from $25\text{ }^\circ\text{C}$ up to $75\text{ }^\circ\text{C}$ prior to insertion into the TEM (this is shown in Fig. 1). Additionally, the samples were monitored during their insertion into the TEM (Supplementary Figure 3) and during removal from the TEM (Supplementary Figure 16).

Additionally, the Ti_3CNT_x and $\text{Mo}_2\text{TiC}_2\text{T}_x$ samples used for *in situ* TEM were measured *ex situ* after annealing within the PPMS. This allowed measurement of the electronic properties down to 10 K (Fig. 2) and measurement of their magnetoresistance (Supplementary Figure 6).

Regarding the measurement of bulk samples *ex situ* to compliment the *in situ* studies, Fig. R6a shows a bulk Ti_3CNT_x sample in its as-prepared state and after annealing at 400 °C in Ar. Similar to the data shown in Fig. 2 of the manuscript, the *ex situ* tested bulk samples also show a semiconductor-like to metallic transition. Additionally, we refer the reviewers to ref. 8, where Kim et al. measured the resistance of $\text{Mo}_2\text{TiC}_2\text{T}_x$ (etched in HF and delaminated with TBAOH) as it was annealed up to 527 °C (shown in Fig. R6b). The sample undergoes a transition from negative dR/dT to positive dR/dT , in agreement with our results. We point out, however, that ref. 8 could not determine the cause of the change in dR/dT owing to a lack of chemical analysis. This work is cited in the manuscript as reference 52.

Fig. R6. a, *Ex situ* measurement of bulk Ti_3CNT_x samples in the as-prepared state and after annealing at 400 °C. The change in dR/dT with annealing above 200 °C is consistent with our *in situ* results. **b**, Data taken from ref. 8 showing that for a bulk $\text{Mo}_2\text{TiC}_2\text{T}_x$ sample, there is a transition in dR/dT with annealing above 500 °C, in agreement with our *in situ* data.

In addition I would like to highlight the following formatting/typographic suggestions:

I found the structure of the manuscript slightly hard to follow – for example the introduction seems to assume significant knowledge of the synthesis of MXenes from MAX phase and that these materials exist as thin (few/many? layer flakes). This is not stated until the results. The summary of the intercalation section reads like a conclusion but is before the discussion of the termination results.

We have added information to the introduction to explain MXene synthesis and general structure (line 70) “MXenes are formed by etching parent MAX compounds to selectively remove the ‘A’ element, *e.g.* Ti_3AlC_2 (layered MAX) \rightarrow $\text{Ti}_3\text{C}_2\text{T}_x$ (2D MXene)^{4,5}. For both device applications and fundamental studies, MXene samples are generally thin films comprised of many flakes, though some studies have focused on monolayer MXene^{6,7}.” We have moved the summary of the intercalation section and the summary of the termination section into a single discussion section at the end of the manuscript, in accordance with Nature Communications guidelines.

Suppl Fig 5 shows Ti_3CNT_x data but refers to TiC_2T_x data (not displayed?)

The position of the $\text{Ti}_3\text{C}_2\text{T}_x$ (002) peak is represented by the vertical dashed line, as stated in the Figure caption. This figure is now in the manuscript, Fig. 2c.

Line 324: Perrson should be Persson?

Line 375 Semiconductor should be semiconductor
Line 251, 253, 446: 'lead' should be 'led'
The references are strongly dominated by self citations.

The typographic errors have been corrected; we thank the reviewer for pointing out these mistakes. Also, we have added relevant publications from other research groups, and we have reduced self-citations.

References for this document

1. Wang, K. *et al.* Fabrication and thermal stability of two-dimensional carbide Ti_3C_2 nanosheets. *Ceram. Int.* **42**, 8419–8424 (2016).
2. Kasap, S. O. *Principles of Electronic Materials and Devices*. (McGraw-Hill, 2006).
3. Kittel, C. *Introduction to Solid State Physics*. (John Wiley & Sons, 2005).
4. Hu, T. *et al.* Chemical Origin of Termination-Functionalized MXenes : $\text{Ti}_3\text{C}_2\text{T}_2$ as a Case Study. *J. Chem. Phys.* **121**, 19254–19261 (2017).
5. Halim, J. *et al.* Variable range hopping and thermally activated transport in molybdenum-based MXenes. *Phys. Rev. B* **98**, 104202 (2018).
6. Alhabeab, M. *et al.* Guidelines for Synthesis and Processing of Two-Dimensional Titanium Carbide ($\text{Ti}_3\text{C}_2\text{T}$). *Chem. Mater.* **29**, 7633–7644 (2017).
7. Sang, X. *et al.* In situ atomistic insight into the growth mechanisms of single layer 2D transition metal carbides. *Nat. Commun.* **9**, 2266 (2018).
8. Kim, H., Anasori, B., Gogotsi, Y. & Alshareef, H. N. Thermoelectric Properties of Two-Dimensional Molybdenum-Based MXenes. *Chem. Mater.* **29**, 6472–6479 (2017).

Reviewer #1 (Remarks to the Author):

The referee appreciates the changes in the manuscript. With the realization that 4-point transport measurements were done the main criticism was satisfactorily answered. Despite the lack of single flake measurements I believe that the revised manuscript provides sufficient new data in a concise review format to attract a broad readership to the field of MAXenes.

I recommend publication.

Reviewer #2 (Remarks to the Author):

The authors provide a thorough response to the comments and I am happy to recommend the paper for publication.

Minor proof errors: line 243- insulting flakes should be insulating and line 415 'in' is missing

REVIEWERS' COMMENTS are in red
Our responses are in black

Reviewer #1 (Remarks to the Author):

The referee appreciates the changes in the manuscript. With the realization that 4-point transport measurements were done the main criticism was satisfactorily answered. Despite the lack of single flake measurements I believe that the revised manuscript provides sufficient new data in a concise review format to attract a broad readership to the field of MAXenes.

I recommend publication.

Reviewer #2 (Remarks to the Author):

The authors provide a thorough response to the comments and I am happy to recommend the paper for publication.

Minor proof errors: line 243- insulting flakes should be insulating and line 415 'in' is missing

We have corrected this error, and we thank the referee for pointing it out.